# Gene selection for optimal prediction of cell position in tissues from single-cell transcriptomics data

Jovan Tanevski[1,2,*], Thin Nguyen[3,*], Buu Truong[4,*], Nikos Karaiskos[5], Mehmet Eren Ahsen[6,7], Xinyu Zhang[8], Chang Shu[8,27], Ke Xu[8], Xiaoyu Liang[8], Ying Hu[9], Hoang VV Pham[4], Li Xiaomei[4], Thuc D Le[4], Adi L Tarca[10], Gaurav Bhatti[11,12], Roberto Romero[11,12], Nestoras Karathanasis[13], Phillipe Loher[13], Yang Chen[14], Zhengqing Ouyang[15], Disheng Mao[16], Yuping Zhang[16], Maryam Zand[17], Jianhua Ruan[17], Christoph Hafemeister[18], Peng Qiu[19,20], Duc Tran[21], Tin Nguyen[21], Attila Gabor[1], Thomas Yu[22], Justin Guinney[22], Enrico Glaab[23], Roland Krause[24], Peter Banda[24], DREAM SCTC Consortium‡, Gustavo Stolovitzky[25], Nikolaus Rajewsky[5,†], Julio Saez-Rodriguez[1,26,†], Pablo Meyer[25]

Single-cell RNA-sequencing (scRNAseq) technologies are rapidly evolving. Although very informative, in standard scRNAseq experiments, the spatial organization of the cells in the tissue of origin is lost. Conversely, spatial RNA-seq technologies designed to maintain cell localization have limited throughput and gene coverage. Mapping scRNAseq to genes with spatial information increases coverage while providing spatial location. However, methods to perform such mapping have not yet been benchmarked. To fill this gap, we organized the DREAM Single-Cell Transcriptomics challenge focused on the spatial reconstruction of cells from the *Drosophila* embryo from scRNAseq data, leveraging as silver standard, genes with in situ hybridization data from the Berkeley *Drosophila* Transcription Network Project reference atlas. The 34 participating teams used diverse algorithms for gene selection and location prediction, while being able to correctly localize clusters of cells. Selection of predictor genes was essential for this task. Predictor genes showed a relatively high expression entropy, high spatial clustering and included prominent developmental genes such as gap and pair-rule genes and tissue markers. Application of the top 10 methods to a zebra fish embryo dataset yielded similar performance and statistical properties of the selected genes than in the *Drosophila* data. This suggests that methods developed in this challenge are able to extract generalizable properties of genes that are useful to accurately reconstruct the spatial arrangement of cells in tissues.

## Introduction

The recent advances in single-cell sequencing technologies have revolutionized the biological sciences. In particular, single-cell RNA-sequencing (scRNAseq) methods allow for transcriptome profiling in a highly parallel manner, resulting in the quantification of thousands of genes across thousands of cells of the same tissue. However, with a few exceptions (1, 2, 3, 4, 5, 6 *Preprint*), current high-throughput scRNAseq methods share the drawback of losing during the cell dissociation step the information about the spatial arrangement of the cells in the tissue.

[1]Institute for Computational Biomedicine, Faculty of Medicine, Heidelberg University Hospital and Heidelberg University, Heidelberg, Germany [2]Department of Knowledge Technologies, Jožef Stefan Institute, Ljubljana, Slovenia [3]Deakin University, Geelong, Australia [4]University of South Australia, Mawson Lakes, Australia [5]Berlin Institute for Medical Systems Biology, Max Delbrück Center for Molecular Medicine in the Helmholtz Association, Berlin, Germany [6]Icahn School of Medicine at Mount Sinai, New York City, NY, USA [7]University of Illinois, Urbana-Champaign, Champaign, IL, USA [8]Department of Psychiatry, Yale School of Medicine, New Haven, CT, USA [9]Center for Biomedical Informatics and Information Technology, National Cancer Institute, Bethesda, MD, USA [10]Department of Obstetrics and Gynecology and Department of Computer Science, Wayne State University, Detroit, MI, USA [11]Perinatology Research Branch, National Institute of Child Health and Human Development (NICHD)/National Insitutes of Health (NIH)/ Department of Health & Human Services (DHHS), Bethesda, MD, USA [12]Perinatology Research Branch, NICHD/NIH/DHHS, Detroit, MI, USA [13]Computational Medicine Center, Thomas Jefferson University, Philadelphia, PA, USA [14]The Jackson Laboratory for Genomic Medicine, Farmington, CT, USA [15]University of Massachusetts, Amherst, MA, USA [16]University of Connecticut, Storrs, CT, USA [17]University of Texas at San Antonio, San Antonio, TX, USA [18]New York Genome Center, New York City, NY, USA [19]Georgia Institute of Technology, Atlanta, GA, USA [20]Emory University, Atlanta, GA, USA [21]University of Nevada, Reno, NV, USA [22]Sage Bionetworks, Seattle, WA, USA [23]Biomedical Data Science Group, Luxembourg Centre for Systems Biomedicine, University of Luxembourg, Esch-sur Alzette, Luxembourg [24]Bioinformatics Core Group, Luxembourg Centre for Systems Biomedicine, University of Luxembourg, Esch-sur Alzette, Luxembourg [25]International Buisness Machines (IBM) T.J. Watson Research Center, Yorktown Heights, NY, USA [26]Joint Research Centre for Computational Biomedicine, Faculty of Medicine, RWTH Aachen University, Aachen, Germany [27]Department of Pediatrics, Columbia University Irving Medical Center, New York, NY, USA

Correspondence: pmeyerr@us.ibm.com
*Jovan Tanevski, Thin Nguyen, and Buu Truong contributed equally to this work
†Nikolaus Rajewsky and Julio Saez-Rodriguez are the senior authors
‡DREAM SCTC Consortium authors and affiliations are listed in the supplementary material

One way of regaining spatial information computationally is to appropriately combine the scRNA dataset at hand with a reference database, or atlas, containing spatial expression patterns for several genes across the tissue. This approach was pursued in a few studies (7, 8, 9, 10, 11). Achim et al (7) identified the location of 139 cells using 72 reference genes with spatial information from whole mount in situ hybridization of a marine annelid and Satija et al (8) developed the *Seurat* algorithm to predict position of 851 zebra fish cells based on their scRNAseq data and spatial information from in situ hybridizations of 47 genes in ZFIN collection (12). In both cases, cell positional predictions stabilized after the inclusion of 30 reference genes. Karaiskos et al (11) reconstructed the early *Drosophila* embryo at single-cell resolution and although the authors were successful in their reconstruction, their work did not lead to a predictive algorithm and mainly focused on maximizing the correlation between scRNAseq data and the expression patterns from in situ hybridizations of 84 mapped genes in the Berkeley *Drosophila* Transcription Network Project (BDTNP). In this project, in situ hybridization data were collected resulting in a quantitative high-resolution gene expression reference atlas (13). Indeed, Karaiskos et al (11) showed that the combinatorial expression of these 84 BDTNP markers sufficed to uniquely classify almost every cell to a position within the embryo.

In the absence of a reference database, it is also possible to regain spatial information computationally solely from the transcriptomics data by leveraging general knowledge about statistical properties of spatially mapped genes against the statistical properties of the scRNA dataset (1, 14). Bageritz et al (1) were able to reconstruct the expression map of a *Drosophila* wing disc using scRNAseq data by correlation analysis. They exploited the coexpression of non-mapping genes to a few mapped genes with known expression patterns, to predict the spatial expression of 824 genes (1). Nitzan et al (14) assumed that cells that are physically close to each other tend to share similar transcription profiles and used the distance between mapping genes in the expression space and cells in the physical space to predict the possible locations of cells based on the distribution of distances between genes in the expression space. Following this approach, they were able to successfully reconstruct the locations of cells of the *Drosophila*, zebra fish embryos, and mammalian tissues from scRNAseq data (14).

Although these approaches are important steps to reconstruct the position of a cell in a tissue from their RNAseq expression, their high performance is conditioned on the selection of informative genes with spatially resolved expression and a global assessment is needed to evaluate the methods used and the number and nature of the best genes required for correctly assigning a location to each cell. With this in mind and to catalyze the development of new methods to predict the location of cells from scRNAseq data, we organized the DREAM Single-cell transcriptomics challenge, which ran from September through November 2018. We set up the challenge with three goals in mind. First, we wanted to foster the design of a variety of algorithms and objectively tested how well they could predict the localization of the cells. Second, we evaluated how the predictive performance of the algorithms was impacted by the number of reference genes with in situ hybridization information included in the predictions. Third, we investigated how the biological information carried in the selected

genes was implemented in the algorithms to determine embryonic patterning.

# Results

## Challenge setup

The challenge, a first of its kind for single-cell data, consisted of predicting the position of 1,297 cells among 3,039 *Drosophila melanogaster* embryonic locations for one half of a stage 6 pre-gastrulation embryo from their scRNAseq data (Fig 1A) (11). At this stage, cells in the embryo are positioned in a single two-dimensional sheet following a bilateral symmetry (left–right), so that only positions in one half of the embryo where considered accounting for the 3,039 locations. Participants used the scRNAseq data for each of the 1,297 cells obtained from the dissociation of 100–200 stage 6 embryos and the spatial expression patterns from in situ hybridizations of 84 genes in the BDTNP database (13). As a source of domain-specific background knowledge that can aid the development of prediction algorithms, we provided information about gene determinants of different tissues such as neuroectoderm, dorsal ectoderm, mesoderm, yolk, and pole cells. We also provided (when available) the regulatory relationship—positive or negative—between the 84 genes in the in situ hybridizations and the rest of the genes. We asked participants to provide an ordered list of 10 most probable locations in the embryo predicted for each of the 1,297 cells using the expression patterns from (i) 60 genes of the 84 in subchallenge 1, (ii) 40 genes out of the 84 in subchallenge 2, and (iii) 20 genes of the 84 in subchallenge 3. The predictions were compared against the best available ground truth location—a silver standard—determined by calculating the maximum correlation using all 84 in situs (11).

DREAM challenges are a platform for crowdsourcing collaborative competitions (15) where a rigorous evaluation of each submitted solution allows for the comparison of their performance. The quality and reproducibility of each provided solution are also ensured. A distinctive feature of this single-cell transcriptomics challenge was the public availability of the entire dataset and the ground truth locations produced by DistMap, a method using the in situ hybridizations available at BDTNP (13), published together with the data (11). We took three actions to mitigate the issue of not having a blinded ground truth. First, for the purpose of predictor gene selection, we allowed the use of scRNAseq data and biological information from other databases but prohibited the use of in situ data. Second, to assess the quality of predictions, we devised three scores (detailed in the Materials and Methods section) that were not disclosed to the participants during the challenge. The scores measured not only the accuracy of the predicted location of the cell but also how well the gene expression in the cell at the predicted location correlates with the expression from the reference atlas, the variance of the predicted locations for each cell, and how well the gene-wise spatial patterns were reconstructed. Finally, we devised a post-challenge cross-validation (CV) scheme to evaluate further soundness and robustness of the methods.

The challenge was organized in two rounds, a leaderboard round and a final round. During the leaderboard round, the participants

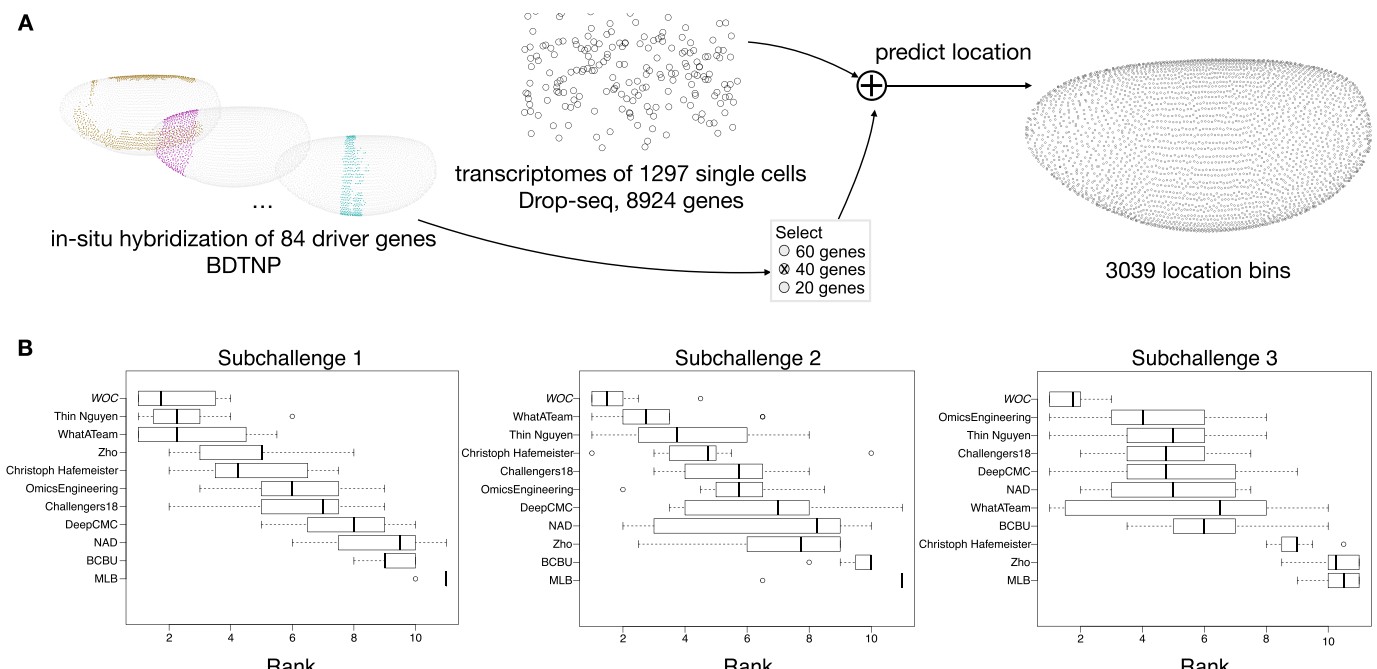

**Figure 1. Overview of the challenge and results.**
**(A)** In the DREAM Single-Cell Transcriptomics challenge, participants were asked to map the location of 1,297 cells to 3,039 location bins of an embryo of *Drosophila melanogaster*, by combining the single-cell RNA-sequencing measurements of 8,924 genes for each cell and the spatial expression patterns from in situ hybridization of 60, 40, or 20 genes, for subchallenge 1, 2, and 3, respectively, for each embryonic location bin, selected from a total of 84 mapped genes. **(B)** Ranking of the top 10 best performing teams and a wisdom of the crowds (WOC) solution, based on results from a post-challenge cross-validated selection and prediction performance measured with three complementary scoring metrics. The boxplots show the distribution of ranks for each team on the 10 test folds. The rank for each fold is calculated as the average of the ranking on each scoring metric.

were able to obtain scores for five submitted solutions before submitting a single solution in the final round. We received submissions from 40 teams in the leaderboard round and 34 submissions in the final round. Of the 34 teams that made submissions in the final round, 29 followed up with public write-ups of their approaches and source code. For subchallenges 1 and 3 we were able to determine a clear best performer, but for subchallenge 2, there were two top ranked teams with statistically indistinguishable difference in performance (see Figs S1–S3).

As stated, given that the ground truth for this challenge was publicly available, we decided to invite the top 10 performing teams to contribute to a post-challenge collaborative analysis phase to assess the soundness and stability of their gene selection and cell location prediction. Consequently, teams were tasked with providing predictions for a 10-fold CV scenario of the *Drosophila* dataset used in the open phase of the challenge. Each team used the same assignment of cells to folds and was evaluated with the challenge scoring approach. To ensure the validity of the findings, we performed all further analysis and interpretation using only the results of the post-challenge phase. In brief, we found that the most frequently selected genes had a relatively high expression entropy, showed high spatial clustering and featured developmental genes such as gap and pair-rule genes in addition to tissue defining markers. We further show that statistical properties of selected genes are robust as they were also identified in an independent scRNAseq dataset used to predict the position of cells in a zebra fish embryo.

## Results overview

When participants had to use 60 or 40 genes for their predictions, in the *Drosophila* subchallenge 1 and 2, respectively, the ranking of the best performing teams in the CV scenario did not change significantly compared with the challenge (Fig 1B cf. Figs S1 and S2). This was not the case in subchallenge 3 as no particular team from the top 10 significantly outperformed the others when using 20 genes for their predictions (Fig S3). Also, the results from the CV show that the approaches generalize well as for all teams the gene selection is performed consistently across the folds (Fig S4) and the SD of the scores is small (Table S1).

For each subchallenge, we combined the gene selection and location predictions from the top 10 participants into a WOC solution (see details below) that performed better compared with the individual solutions (Fig 1B). For comparison, the scores obtained by the best performing teams and the WOC solution are shown in Table 1.

Regarding the approaches used to solve the challenge, there was more diversity in the methods for gene selection than for location prediction. For the latter, the most used one was unsupervised or supervised feature importance estimation and ranking. For example, in a supervised feature importance estimation approach, a Random Forest (BCBU, OmicsEngineering) or a neural network (DeepCMC (16 Preprint), NAD) were trained to predict the coordinates of each cell, given the transcriptomics data as input and using either all genes or the genes with available in situ hybridization

**Table 1. Best mean score for metrics *s1*, *s2*, and *s3* achieved by the teams (Thin Nguyen, WhatATeam, and OmicsEngineering) and the WOC solution.**

|  | s1 | | s2 | | s3 | |
|---|---|---|---|---|---|---|
|  | Teams | WOC | Teams | WOC | Teams | WOC |
| Subchallenge 1 | 0.76 (±0.04) | 0.73 (±0.04) | 2.52 (±0.28) | 2.16 (±0.20) | 0.59 (±0.01) | 0.62 (±0.01) |
| Subchallenge 2 | 0.69 (±0.03) | 0.70 (±0.05) | 1.16 (±0.12) | 1.84 (±0.26) | 0.67 (±0.02) | 0.65 (±0.01) |
| Subchallenge 3 | 0.65 (±0.05) | 0.68 (±0.03) | 0.88 (±0.13) | 1.42 (±0.16) | 0.79 (±0.02) | 0.71 (±0.01) |

The SD of scores across folds is in parenthesis. *s1* measures how well the expression of the cell at the predicted location correlates to the expression from the reference atlas and includes the variance of the predicted locations for each cell, *s2* measures the accuracy of the predicted location, and *s3* measures how well the gene-wise spatial patterns were reconstructed. For more details on the scoring metrics, see the Materials and Methods section.

measurements. There were examples of unsupervised feature importance estimation and ranking by expression-based clustering (NAD, Christoph Hafemeister, MLB), or a greedy feature selection based on predictability of expression from other genes (WhatATeam). A small number of teams (WhatATeam and NAD) used background knowledge about location specific marker genes, or the expected number of location clusters, to inform the gene selection.

Two types of approaches were taken to predict cell location, the most frequent one being a similarity-based prediction, such as the maximum Matthews correlation coefficient (MCC) between the binarized transcriptomics and the in situs that was proposed by Karaiskos et al (11) and used to obtain the silver standard. Another well performing approach was combining the predictions of a machine learning model and MCC. In this scheme, models were trained to predict either the coordinates of each cell or the binarized values of the selected in situs, given transcriptomics data as input. The predictions were then made by selecting the location bins that corresponded to the nearest neighbors of the predicted values. The high ranking of the teams that used these two classes of approaches show that the selection of genes for which in situ measurements are available is essential (See summary of methods in Tables S2 and S3 as well as links to the write-ups and the code provided by each team Table S4). In addition, given the high diversity of approaches to gene selection, we focused our analysis on better understanding the properties of frequently selected genes to provide recommendations for future experimental designs.

To confirm that the *Drosophila* findings were robust, we included in our analysis an additional dataset consisting of scRNAseq measurements of 851 cells from a zebra fish embryo previously considered by Satija et al (8) together with spatial information for 64 locations from in situ hybridizations of 47 landmark genes from the ZFIN collection (12). As ground truth for predicting the position of the zebra fish embryo cells, we used the location predictions produced by applying DistMap, choosing a threshold for scRNAseq expression to maximize the MCC with all 47 in situ genes. The method proposed by Satija et al (8) can also be used to produce ground truth locations, but for consistency with the challenge we decided to use DistMap. Note that we compared the predicted locations and the redundancy that arises from trying to place 851 cells in 64 different locations when using DistMap and *Seurat*. The results in Fig S5 show notably that on average DistMap and *Seurat* agree for 5 of 10 positions for a cell, with only 46 cells showing no agreement between the methods when assigning 64 possible locations. This represents a high level of overall agreement and enhances the confidence in the silver standard. We tasked once again the top 10 teams to select 20 and 40 genes to place correctly the zebra fish cells in a 10-fold CV scheme. Team ranks varied on this additional dataset (Figs S6 and S7), but the scores achieved were of the same order as in *Drosophila* (Tables S1 and S5), and we found that the selected genes had similar statistical properties (Fig S8 and Tables 2 and S6).

### Combining selected genes

The selection of a smaller subset of in situs used for cell location prediction was the hallmark that differentiated the subchallenges. As it is combinatorially unfeasible to evaluate all sets of 20, 40, or 60 genes from the 84 available, the top 10 ranked teams selected genes mostly based on feature ranking algorithms using normalized transcriptomics data (for more details see Table S3). For prediction robustness and biological relevance purposes, one would expect that genes would be consistently identified across the folds of the 10-fold CV scheme. All the more as the correlation of scRNAseq expression for all pairs of 84 mapped genes across cells was low (Fig S9). Indeed, for all subchallenges, the 10 methods led to a significant and consistent selection of genes across folds, even as we measured higher variance and lower similarity as the number of selected genes decreased (see Fig S4). The consistency of gene selection across folds was also confirmed by the results from the analysis of the zebrafish embryo (see Fig S10).

For each subchallenge, we counted the number of times that the genes were selected by all teams in all folds and observed that a high proportion of genes are consistently selected across subchallenges (Fig 2A and B). 40% of the top 20, 67% of the top 40, and 81% of the top 60 most frequently selected genes are the same for all three subchallenges (Fig 2B). The ranks assigned to all genes in the three subchallenges were also highly correlated. Namely, the rank correlations range from 0.69 between subchallenges 1/3, to 0.83 between subchallenges 1/2 and 2/3, also shown when measured using the Jaccard similarity of the sets of top-k most frequently selected genes for pairs of subchallenges (Fig 2C). The lists of most frequently selected 60, 40, and 20 genes in subchallenges 1, 2, and 3, respectively, are available in the Supplemental Data 1 (Table S7). To validate the predictive power of the most frequently selected genes, we established the cell locations using DistMap and found that using those genes gave significantly better results than a random selection of genes (Fig 2D and see below WOC section for more details). We conclude that the gene selection is not only consistent by team across folds, but also across teams and subchallenges.

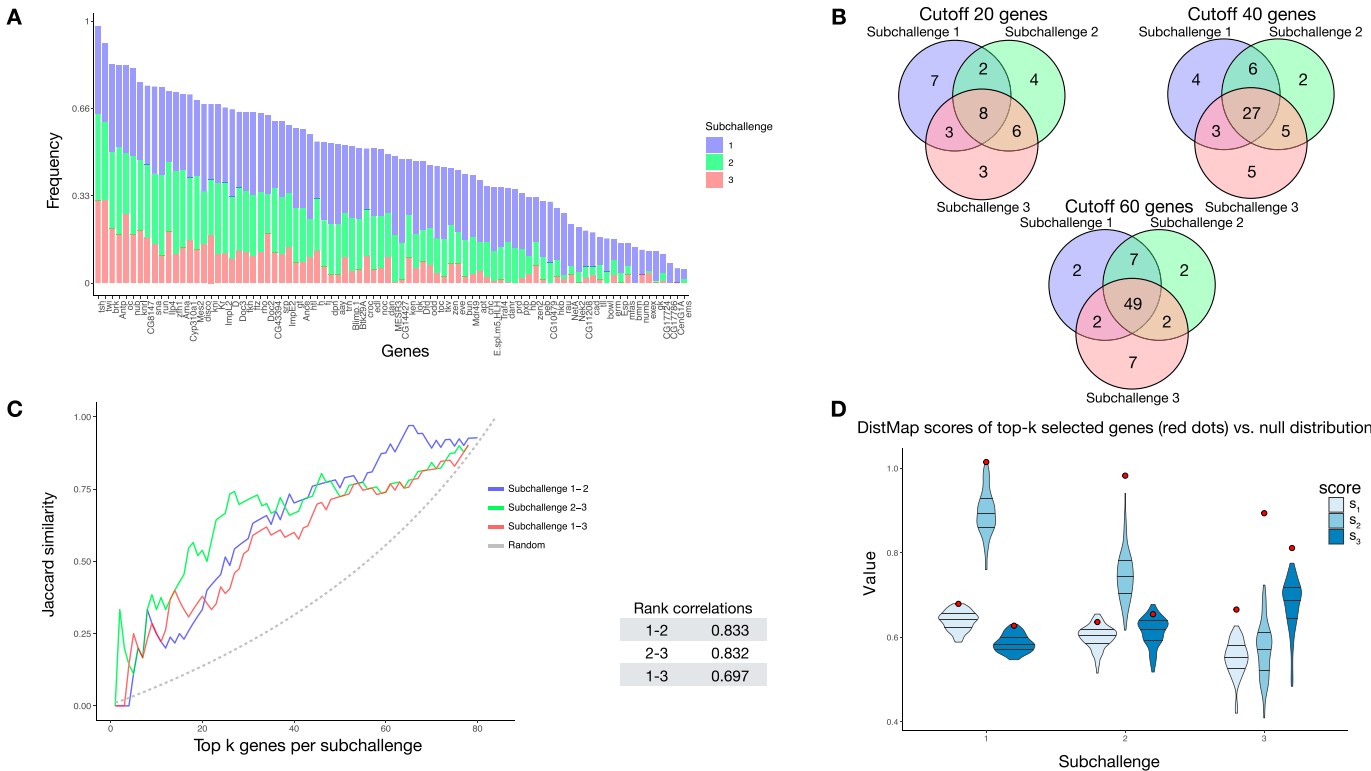

**Figure 2. Analysis of gene selection.**
The results in all figures were generated from the genes that were selected by the top performing teams in the post-challenge cross-validation scenario. **(A)** Frequency of selected genes in subchallenge 1 (blue), subchallenge 2 (green), and subchallenge 3 (red). The genes are ordered according to their cumulative frequency. **(B)** Venn diagrams of the most frequently selected genes in the subchallenges with cutoff at 20, 40, and 60 most frequently selected genes, corresponding to the number of genes required for each subchallenge. **(C)** Left: the similarity of most frequently selected genes for pairs of subchallenges. The Jaccard similarity measures $|A \cap B|$ the ratio of the size of the intersection and the union of two sets $J(A, B) = |A \cup B|$. Right: table of correlations between gene rankings (by frequency) for pairs of subchallenges. **(D)** Validation of the performances of the most frequently selected 60, 40, and 20 genes in the respective subchallenges, also used as the wisdom of the crowds (WOC) selection of genes. The violin plots represent null distribution of scores obtained by 100 randomly selected sets of 60, 40, and 20 genes using DistMap. The red dots represent the performance obtained by using DistMap with the most frequently selected genes equivalent to the WOC selection of genes.

## Properties of frequently selected genes

We conjectured that the most frequently selected genes should carry enough information content collectively to uniquely encode a cell's location. Furthermore, genes should also contain location specific information, that is, their expression should cluster well in space.

To quantify these features, we calculated the entropy and the join count statistic for spatial autocorrelation of the in situs (see the Materials and Methods section for description). We observed that most of the in situ genes have relatively high entropy as observed by the high density in the upper part of the plots and show high spatial clustering, that is, show values of the join count test statistic lower than zero (see Fig 3A).

To test our conjectures of high entropy and spatial correlation, we tested the significance for the shift of the values between the most frequently selected genes and the non-selected genes from all in situs. We observed for all subchallenges a significant value shift for the autocorrelation statistic as evaluated by a one sided Mann–Whitney U test (see bottom of Fig 3A). Although we see a decrease of the statistical significance of the mean value shift for the distribution of entropy values of the selected subsets of genes,

the shift is significant for all subchallenges. At the same time, we observed that the tail of the distribution shortens.

To test whether the information relative to different cell types is retained with the selected subset of 60, 40, or 20 most frequently selected genes, we embedded the cells into a 2D space using t-distributed stochastic embedding (t-SNE) (17) (Figs 3B and S11). Notably, we found that the nine prominent cell clusters identified in the study by Karaiskos et al (11), while using the whole scRNAseq expression dataset, are preserved in our t-SNE embedding and clustering experiments when considering only the most frequently in situ selected 60 or 40 genes from subchallenges 1 and 2. This is not the case for subchallenge 3 as the number of cell clusters is reduced when considering the most frequently selected 20 genes.

We then analyzed the differentially expressed genes for each of the nine clusters in all subchallenges. In particular, we focused on the frequently selected mapped genes to discover representative genes identifying each cluster. The results from the one-versus-all clusters differential expression analysis for each subchallenge are presented in Figs S12–S14. We find that for each subchallenge, 81, 82, and 77 from the 84 mapped genes are significantly differentially expressed in at least one of the clusters, respectively (Fig 4). To assign genes to each cluster identity, we first selected the top three

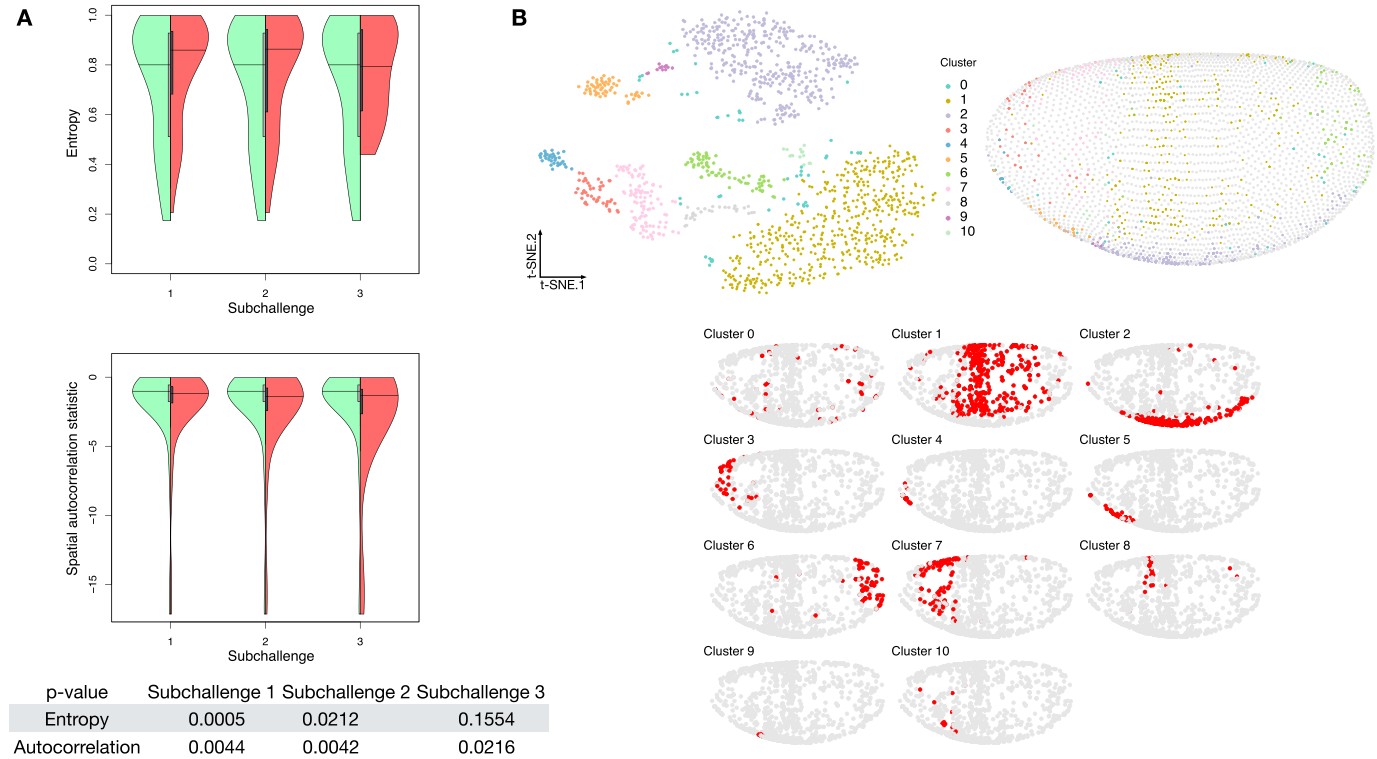

**Figure 3. Properties of selected genes.**
**(A)** Double violin plots of the distribution of entropy and spatial autocorrelation statistic of (left, green) all in situs calculated on all embryonic location bins and (right, red) the most frequently selected 60, 40, and 20 genes in the respective subchallenges. Bottom table: *P*-values of a one-sided Mann–Whitney U test of location shift comparing the selected (red part of the violin plot) genes versus the non-selected genes (green part of the violin plot). **(B)** Top left: visualization of the transcriptomics data containing only the most frequently selected 60 genes from subchallenge 1 by the top-performing teams (embedding to 2D by t-SNE). Each point (cell) is filled with the color of the cluster that it belongs to (density-based clustering with DBSCAN). Top right: spatial mapping of the cells in the *Drosophila* embryo as assigned by DistMap using only the 60 most frequently selected genes from subchallenge 1. The color of each point corresponds to the color of the cluster from the t-SNE visualization. Bottom: highlighted (red) location mapping of cells in the *Drosophila* embryo for each cluster separately.

differentially expressed genes in each cluster and in each sub-challenge. We obtained a representative set of 23, 22, and 13 mapped genes for each subchallenge, which shows that the participants selected a diverse set of most differentially expressed genes representative of various spatial locations. The intersection of these sets contains 11 genes from which 10 are among the top 20 most selected genes (Fig 4 bottom, Figs S15 and S16 for remaining genes for each subchallenge). Thus, in conclusion, participants' preferred genes are also mostly differentially expressed between clusters of the scRNAseq data.

Next, we aimed to discover other statistical properties of the transcriptomics data that might inform future experimental designs when selecting target genes for in situ hybridization. We associated their statistical properties, such as variance of gene expression $\sigma^2$ across cells, the coefficient of variation $c_v = \frac{\sigma}{\mu}$, the number of cells with expression 0 and the entropy of binarized expression $H_b$, to those in the in situs that were found to be indicative of good performance, that is, entropy $H$ and the value of the join count statistic $Z$. We calculated these statistical features across cells for the subset of 84 genes from the transcriptomics data for which we also have in situ measurements, and then calculated the correlation across genes for each of these metrics and the measured spatial properties of interest (see Table 2).

Although the selection of highly variable genes was one of the approaches used by some of the top 10 teams, the variance for each gene in the scRNAseq expression was less correlated than other properties to the entropy of the corresponding in situ measurements of that gene. We observed that the positive correlation of the entropy to the variance of each gene becomes negative when calculated against their coefficient of variation. This negative correlation can have two sources, the genes with high entropy may have low SD or high mean expression. Because the entropy is positively correlated with the variance of expression, we can conclude that the negative correlation is a result of highly expressed genes. This makes sense as a known drawback of scRNAseq is a high number of dropout events for lowly expressed genes (18). An observation that is further supported by the negative correlation of the entropy and the number of cells with zero expression. Finally, the highest correlation of in situ entropy was to the entropy of the binarized expression. Regarding the spatial autocorrelation, all statistical features of the transcriptomics were only slightly positively correlated to the join count statistic except for the entropy of binarized expression which had negative correlation.

We confirmed the results from the correlation of the properties of the in situs to the statistical properties of the gene expression in

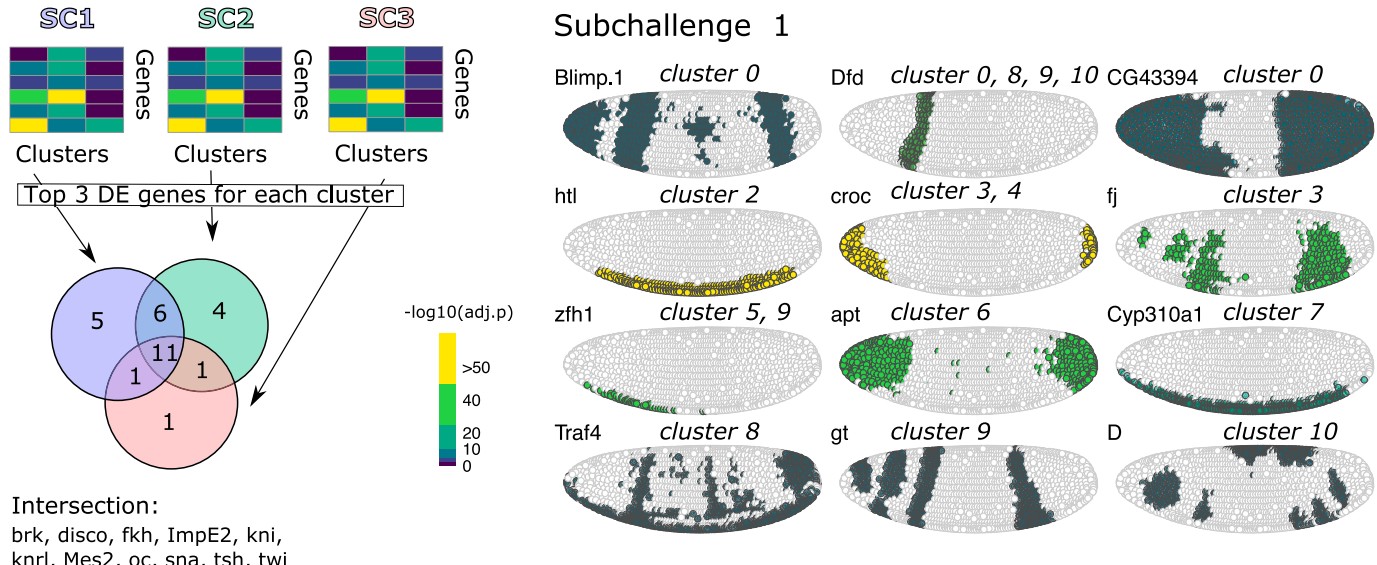

**Figure 4. Identifying cluster-specific differentially expressed genes.**
Top left: For each subchallenge, the top three differentially expressed from the set of 60, 40, and 20 most frequently selected genes for each cluster were used to identify common and subchallenge-specific representative genes. Bottom left: the intersection of the sets of representative genes for each subchallenge contains 11 common genes. Right: examples of expression of remaining genes for subchallenge 1 are shown as an illustration of how they can be used to identify specific clusters.

the transcriptomics on the zebra fish dataset (see Table S6). Taken together, our findings suggest that high expression, differential expression, high entropy, and spatial clustering of the binarized expression are indicative of informative mapping genes and should guide future experimental designs.

## Combining location predictions

A recurrent observation across DREAM challenges is that an ensemble of individual predictions usually performs better and is more robust than any individual method (19, 20). This phenomenon, also common in other contexts, is denoted as the wisdom of the crowds (WOC) (15). In a typical challenge, individual methods output a single probability reflecting the likelihood of occurrence of an event. The WOC prediction is then constructed in an unsupervised manner by averaging the predictions of individual methods. In the single-cell transcriptomics challenge, we leveraged the diversity of

**Table 2. Correlations of transcriptomics to in situ properties of the genes where both measurements are available.**

| | | In situ | | |
|---|---|---|---|---|
| | | Correlation | H | Z |
| scRNASeq | $\sigma^2$ | | 0.5 | 0.18 |
| | CV | | −0.69 | 0.26 |
| | 0 | | −0.64 | 0.29 |
| | $H_b$ | | 0.72 | −0.3 |

$\sigma^2$, variance of a gene across cells; CV, coefficient of variation; 0, number of cells with zero expression; $H_b$, entropy of binarized expression; H, entropy; Z, join count test statistic.

the top performing methods for location prediction only (Table S2) to construct a WOC ensemble prediction. The WOC location prediction approach does not take the genes used by the teams to make the predictions into account. However, after the WOC predictions are generated and to score them, we used the most frequently selected genes for every subchallenge (see below). Given that in the scRNAseq prediction challenge, participants had to submit 10 positions per cell, we developed a novel method based on k-means clustering to generate the WOC predictions. Fig 5 displays a diagram of the k-means approach, where, for each single cell, we first used k-means clustering to group the locations predicted by the individual teams (21) using the Euclidean distance between the locations as the metric. To find the optimal $k$, we used the elbow method, that is, we chose a $k$ that saturates the sum of squares between clusters (22). Note that each cluster consists of a group of locations and each location is predicted by one or more teams. Hence, for each cluster, we calculated the average frequency that its constituent locations are predicted by individual teams. We then picked the cluster with the highest average frequency and ranked each location in this cluster based on how frequently it was predicted by individual methods. For each cell, the final prediction of the proposed WOC method consisted of the top 10 locations based on the above ranking. The k-means approach is based on the intuition that a single cell belongs to one location, and its expression is mostly similar to that of cells in locations surrounding it. The results show that the proposed WOC solution performed better than the individual solutions (Fig 1B). In particular, as shown in Table 1, the superior performance of the WOC approach can be attributed to improvements in scores $s_1$ and $s_2$, that is, the correlation of expression of the cells at their predicted locations with the reference atlas, the accuracy, and low variance of the predicted most probable locations to the ground truth location for each cell.

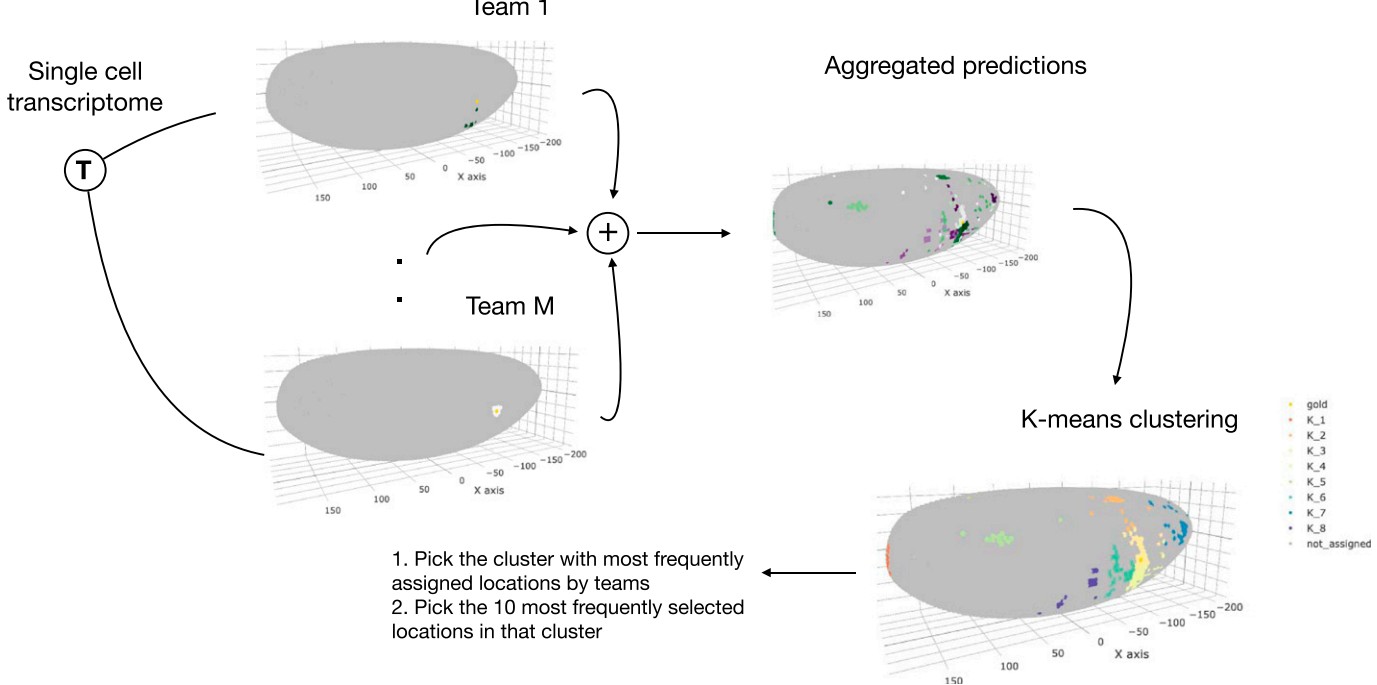

**Figure 5. Wisdom of crowds location prediction.**
The location predictions for each cell by the top performing teams in the post-challenge cross-validation phase were aggregated in the wisdom of the crowds solution based on a k-means clustering approach.

### Validation of frequently selected genes

We defined a simple procedure to obtain a WOC gene selection for each of the subchallenges. It consisted of selecting the most frequently selected genes for each subchallenge (different colored bars in Fig 2A). To validate the predictive performance of the WOC gene selection independent from the participant's location prediction methods, we predicted the cell locations using DistMap, the method used to generate the ground truth locations for each cell for the challenge. We scored the predictions using the same scoring metrics as for the challenge, estimating the significance of the scores through generated null distributions of scores for each subchallenge. The null distribution of the scores was generated by scoring the DistMap location prediction using 100 different sets of randomly selected genes. For each subchallenge and each score, we estimated the empirical distribution function and then calculated the percentile of the values of the scores obtained with the WOC gene selection.

The null distributions and the values of the scores obtained with the WOC gene selection are shown in Fig 2D. All values of the scores for subchallenge 1 fall in the 99th percentile. For subchallenge 2, $s_1$ and $s_3$ fall into the 92nd percentile and $s_2$ in the 100th percentile. For subchallenge 3, all scores fall in the 100th percentile. Overall, the performance of DistMap with the WOC selected genes performs significantly better than a random selection of genes. The actual values of the scores are on par with those achieved by the top 10 teams in the challenge.

## Discussion

In this article, we report the results of a crowdsourcing effort organized as a DREAM challenge (15) to predict the spatial arrangement of cells in a tissue from their scRNAseq data. Analysis of the top performing methods provided many unbiased insights such as the usage of either similarity-based approaches or machine learning models to predict cell location. The latter, in accordance with current literature (8, 11, 14), were shown to be preferable. We do not think this is due to bias induced by the fact that the silver standard was generated using a similarity-based approach.

Indeed, we showed for the zebra fish dataset that the silver standard is robust to the usage of *Seurat* (Fig S5), a different method to generate the cells' positions. Also, the good performance and robustness of nonlinear machine learning methods (Table S1) is proof that the association between the expression of mapped genes and a cell's position is not due to a simple gradient of expression in space. Consequently, we conjecture that a combination of these two approaches would be most preferable for predicting cells' unknown locations. Namely, similarity-based approaches can be used to make position assignments for a subset of cells with high similarity of gene expression to a spatially resolved reference. Then, machine learning approaches take advantage of this information to predict the positions of the remaining cells.

Given that for all approaches, the selection of informative genes with spatially resolved expression is essential, the main finding of this study is how to select these genes based on their cell-to-cell expression variability in the *Drosophila* and zebra fish embryos to best predict a cell's localization. The most selected genes had a relatively high entropy, hence high variance and high expression values while also showing high spatial clustering. The smaller the number of selected genes, that is, going from 60 to 40 and to 20, the more these features became apparent (Figs 3 and S8). The observed advantage of genes with high overall expression in cells might lead

to less dropout counts in the scRNAseq data, a known disadvantage of the technology, leading to more accuracy in the cell placement. However, we also found that most in situ genes were differentially expressed across cell clusters in the scRNAseq data and top three differentially expressed genes have notable overlap across challenges (see Figs 4 and S15). For *Drosophila*, the nine prominently spatially distinct cell clusters previously identified (11) are preserved when considering the most frequently selected 60 or 40 genes and for 40 genes in zebra fish. However, for both organisms, the number of clusters is reduced when considering only the most frequently selected 20 genes. This finding is in line with the conclusions of Howe et al (12), where in a related task of location prediction, the performance stabilized after the inclusion of 30 genes. Finally, the WOC gene selection and the k-means clustered WOC model for cell localization performed comparably or better than the participant's models, showing once more the advantage of the wisdom of the crowds. All these results can be explored for *Drosophila* in animated form at https://dream-sctc.uni.lu/.

Given that it has been shown that positional information of the anterior–posterior (A-P) axis is encoded as early in the embryonic development as when the expression of the gap genes occurs (23, 24), we thought that it should be possible to implement in algorithms for this challenge the information contained in the regulatory networks of *Drosophila* development (25). Although only a small number of participants—including the best performers—directly used biological information related to the regulation of the genes or their connectivity, the most frequently selected genes in all three subchallenges have interesting biological properties. Indeed, gap genes such as *giant* (*gt*), *kruppel* (*kr*), and *knirps* (*kni*) were selected in all three subchallenges (see Fig S17 and Table S7 that also includes *kni*-like *knrl*), although *tailless* (*tll*) and *hunchback* (*hb*) were not. Along the A-P axis, maternally provided *bicoid* (*bcd*) and *caudal* (*cad*) first establish the expression patterns of gap and terminal class factors, such as *hb*, *gt*, *kr*, and *kni*. These A-P early regulators then collectively direct transcription of A-P pair-rule factors, such as *even-skipped* (*eve*), *fushi-tarazu* (*ftz*), *hairy* (*h*), *odd skipped*, (*odd*), paired (*prd*), and runt (*run*) which in turn cross-regulate each other. Not being part of the in situs, neither *bcd*, nor *cad* were selected but *ama* sitting near *bcd* in the genome might have been selected for its similar expression properties. Furthermore, we also found that pair-rule genes were most prominently selected in subchallenges 1 (*eve*, *odd*, the paired-like *prd* and *bcd*) and 2 (*h*, *ftz* and *run*). A similar cascade of maternal and zygotic factors controls patterning along the dorsal–ventral axis were *dorsal* (*d*), *snail* (*sna*), and *twist* (*twi*) specify mesoderm and the pair-rule factors *eve* and *ftz* specify location along the trunk of the A-P axis. Again, *sna* and *twi* were selected in all subchallenges and *d* in subchallenges 1 and 2. These selected transcription factors specify distinct developmental fates and can act via different cis-regulatory modules, but their quantitative differences in relative levels of binding to shared targets correlate with their known biological and transcriptional regulatory specificities (26). The rest of the selected genes were the homeobox genes (*nub* and *antp*) and differentiators of tissue such as mesoderm (*ama*, *mes2*, and *zfh1*), ectoderm (*doc2* and *doc3*), neural tissue (*noc*, *oc*, and *rho*), and EGFR pathway (*rho* and *edl*). The observation that gap and pair-rule genes were prominently

selected is notable as it shows that information providing the correct localization of a cell is encoded in scRNAseq at such early developmental stages. Previous publications (23, 24) had shown that the four gap genes could precisely place dorsal position for cells, but the results described herein go beyond and show that cells can be placed in the 3-D embryo map. The complete lists of most frequently selected genes are available in Table S7.

Because only a publicly available silver standard existed, the organization of this DREAM challenge brought risks. Without the post-challenge phase, it would have been impossible to ensure that the approach and methods implemented were robust and sound. Overall, the single-cell transcriptomics challenge unveils not only the best gene selection methods and prediction approaches to localize a cell in the *Drosophila* and zebra fish embryo but also explains the biological and statistical properties of the genes selected for the predictions, including that spatially auto-correlated genes are the most informative (1, 14). However, we think that the approach defined here could be used or adapted when performing similar cell-placing tasks in other organisms, including human tissues. In fact, for all organisms studied, selecting the appropriate marker genes for optimal cartography has been shown to have a large effect on the performance (14). Given the importance of spatial arrangements for disease development and treatment, we foresee an application of these methods to medical questions as well.

# Materials and Methods

### Scoring

We scored the submissions for the three subchallenges using three metrics $s_1$, $s_2$, and $s_3$. $s_1$ measured how well the expression of the cell at the predicted location correlates to the expression from the reference atlas and included the variance of the predicted locations for each cell, whereas $s_2$ measured the accuracy of the predicted location and $s_3$ measured how well the gene-wise spatial patterns were reconstructed.

Let $c$ represent the index of a cell, given in the transcriptomics data in the challenge where $1 \leq c \leq 1{,}297$. Each cell $c$ is located in a bin $\varepsilon_c \in \{1...3{,}039\}$ at a position with coordinates $r(\varepsilon_c) = (x_c, y_c, z_c)$. Each cell is associated with a binarized expression profile $t_c = (t_{c1}, t_{c2},...,t_{cE})$, where $1 \leq E \leq 8{,}924$, and a corresponding binarized in situ profile $f_c = (f_{c1}, f_{c2},...,f_{cK})$, where the maximum possible value of $K$ for which we have in situ information is $K = 84$. For different subchallenges, we consider $K \in \{20, 40, 60\}$. Using $K$ selected genes, the participants were asked to provide an ordered list of 10 most probable locations for each cell. We represent with the mapping function $A(c, i, K)$ the value of the predicted $i$-th most probable location for cell $c$ using $K$ in situs.

For the first scoring metric $s_1$, we calculated the weighted average of the MCC between the in situ profile of the ground truth cell location $f_{\varepsilon_c}$ and the in situ profile of the most probable predicted location for that cell.

$$s_1 = \sum_{c=1}^{N} \frac{p_K(c, A)}{\sum_{i=1}^{N} p_K(i, A)} MCC\left(f_{A(c, 1, K)}, f_{\varepsilon_c}\right),$$

where $N$ is the total number of cells with predicted locations.

The MCC, or $\phi$ coefficient, is calculated from the contingency table obtained by correlating two binary vectors. The MCC is weighted by the inverse of the distance of the predicted most probable locations to the ground truth location $p_K(c)$. The weights are calculated as $p_K(c, A) = \frac{\widetilde{d_{84}(c,A)}}{d_K(c,A)}$, where $d_K(c, A) = \frac{1}{10}\sum_{i=1}^{10}\|r(A(c, i, K)) - r(\varepsilon_c)\|_2$, $\widetilde{d_{84}(c, A)}$ is the value of $d_K(c, A)$ using the ground truth most probable locations assigned with $K = 84$ using DistMap, and $\|\cdot\|_2$ is the Euclidean norm.

The second metric $s_2$ is simply the average inverse distance of the predicted most probable locations to the ground truth location.

$$s_2 = \frac{1}{N}\sum_{c=1}^{N} p_K(c, A).$$

Finally, the third metric $s_3$ measures the accuracy of reconstructed gene-wise spatial patterns.

$$s_3 = \sum_{s=1}^{K} \frac{MCC\left(t_{cs}, f_{\varepsilon_c s}\right)\forall_c}{\sum_{i=1}^{K} MCC\left(t_{ci}, f_{\varepsilon_i r}\right)\forall_c} MCC\left(t_{cs}, f_{A(c,1,K)s}\right)\forall_c,$$

where $\forall_c$ denotes that the *MCC* is calculated cell wise for each gene.

For 287 of the 1,297 cells, the ground truth location predictions were ambiguous, that is, the MCC scores were identical for multiple locations. These cells were removed both from the ground truth and the submissions before calculating the scores.

The teams were ranked according to each score independently. The final assigned rank $r_t$ for team $t$ was calculated as the average rank across scores. Teams were ranked based on the performance as measured by the three scores on 1,000 bootstrap replicates of the submitted solutions. The three scores were calculated for each bootstrap. The teams were then ranked according to each score. These ranks were then averaged to obtain a final rank for each team on that bootstrap. The winner for each subchallenge was the team that achieved the lowest ranks. We calculated the Bayes factor of the bootstrap ranks for the top performing teams. Bayesian factor of three or more was considered as a significantly better performance. The Bayes factor of the 1,000 bootstrapped ranks of teams $T_1$ and $T_2$ was calculated as follows:

$$BF(T_1, T_2) = \frac{\sum_{i=1}^{1000}\mathbf{1}\left(r(T_1)_i < r(T_2)_i\right)}{\sum_{i=1}^{1000}\mathbf{1}\left(r(T_1)_i > r(T_2)_i\right)},$$

where $r(T_1)_i$ is the rank of team $T_1$ on the $i$-th bootstrap, $r(T_2)_i$ is the rank of team $T_2$ on the $i$-th bootstrap, and $\mathbf{1}$ is the indicator function.

### Entropy and spatial autocorrelation

The entropy of a binarized in situ measurements of gene $G$ was calculated as follows:

$$H(G) = -p\log_2 p - (1 - p)\log_2(1 - p),$$

where p is the probability of gene $G$ to have value 1. In other words, $p$ is the fraction of cells where $G$ is expressed.

The join count statistic is a measure of a spatial autocorrelation of a binary variable. We will refer to the binary expression 1 and 0 as black ($B$)

and white ($W$). Let $n_B$ be the number of bins where $G$ is expressed ($G = B$), and $n_W = n - n_B$ the number of bins where $G$ is not expressed ($G = W$). Two neighboring spatial bins can form join of type $J \in \{WW, BB, BW\}$.

We are interested in the distribution of BW joins. If a gene has a lower number of BW joins that the expected number of BW, then the gene is positively spatially auto-correlated, that is, the gene is highly clustered. Contrarily, higher number of BW joins points toward negative spatial correlation, that is, dispersion.

Following Cliff and Ord ([27]) and Sokal and Oden ([28]), the expected count of BW joins is as follows:

$$\mathbb{E}[BW] = \frac{1}{2}\sum_i\sum_j\frac{w_{ij}n_B^2}{n^2},$$

where the spatial connectivity matrix $w$ is defined as follows:

$$w_{ij} = \begin{cases} 1 & \text{if } i \neq j \text{ and } j \text{ is in the list of 10 nearest neighbors of } i \\ 0 & \text{otherwise} \end{cases}$$

The variance of BW joins is as follows:

$$\sigma_{BW}^2 = \mathbb{E}[BW^2] - \mathbb{E}[BW]^2.$$

where the term $\mathbb{E}[BW^2]$ is calculated as follows:

$$\mathbb{E}[BW^2] = \frac{1}{4}\left(\frac{2x_2 n_B n_W}{n^2} + \frac{(x_3 - 2x_2)n_B n_W(n_B + n_W - 2)}{n^3} + \frac{4(x_1^2 + x_2 - x_3)n_B^2 n_W^2}{n^4}\right),$$

where $x_1 = \sum_i\sum_j w_{ij}$, $x_2 = \frac{1}{2}\sum_i\sum_j\left(w_{ij} - w_{ij}\right)^2$, $x_3 = \sum_i\left(\sum_j w_{ij} + \sum_j w_{ij}\right)^2$.

Note that the connectivity matrix w can also be asymmetric because it is defined by the nearest neighbor function.

Finally, the observed BW counts are follows:

$$BW = \frac{1}{2}\sum_i\sum_j w_{ij}(G_i - G_j)^2.$$

The join count test statistic is then defined as follows:

$$Z(BW) = \frac{BW - \mathbb{E}[BW]}{\sqrt{\sigma_{BW}^2}},$$

which is assumed to be asymptotically normally distributed under the null hypothesis of no spatial autocorrelation. Negative values of the $Z$ statistic represent positive spatial autocorrelation, or clustering, of gene $G$. Positive values of the $Z$ statistic represent negative spatial autocorrelation, or dispersion, of gene $G$.

### Implementation details

The challenge scoring was implemented and run in R version 3.5, the post-analysis was performed with R version 3.6 and the core tidyverse packages. We used the publicly available implementation of DistMap (https://github.com/rajewsky-lab/distmap). MCC calculated with R package mccr (0.4.4). t-SNE embedding and visualization produced with R package Rtsne (0.15). DBSCAN clustering with R

package dbscan (1.1-4). We used t-SNE aiming for high accuracy ($\theta$ = 0.01), then clustered the t-SNE embedded data using density-based spatial clustering of applications with noise (DBSCAN) (29). DBSCAN determines the number of clusters in the data automatically based on the density of points in space. The minimum number of cells in a local neighborhood was set to 10 and the parameter $\varepsilon$ = 3.5 was selected by determining the elbow point in a plot of sorted distances of each cell to its 10th nearest neighbor.

### Code availability

Scoring scripts for the challenge are available at https://github.com/dream-sctc/Scoring. *Drosophila* and zebra fish 10-fold CV datasets can be found at https://github.com/dream-sctc/Data.

## Data description

### Reference database

The reference database comes from the BDTNP. The in situ expression of 84 genes (columns) is quantified across the 3,039 *Drosophila* embryonic locations (rows) for raw data and for binarized data. The 84 genes were binarized by manually choosing thresholds for each gene.

### Spatial coordinates

One half of *Drosophila* embryo has 3,039 cells places as x, y, and z (columns) for a total of 3,039 embryo locations (rows) and a total of 3,039 3 coordinates.

### scRNAseq

The scRNAseq data are provided as a matrix with 8,924 genes as rows and 1,297 cells as columns. In the raw version of the matrix, the entries are the raw unique gene counts (quantified by using unique molecular identifiers). The normalized version is obtained by dividing each entry by the total number of unique molecular identifiers for that cell, adding a pseudocount and taking the logarithm of that. All entries are finally multiplied by a constant. For a given gene, and only considering the Drop-seq cells expressing it, we computed a quantile value above (below) which the gene would be designated ON (OFF). We sampled a series of quantile values and each time the gene correlation matrix based on this binarized version of normalized data versus the binarized BDTNP atlas was computed and compared by calculating the mean square root error between the elements of the lower triangular matrices. Eventually, the quantile value 0.23 was selected, as it was found to minimize the distance between the two correlation matrices. The short sequences for each of the 1,297 cells in the raw and normalized data are the cell barcodes.

## Materials and correspondence

Requests for data, resources, and or reagents should be directed to Pablo Meyer (pmeyerr@us.ibm.com).

# Supplementary Information

# Acknowledgements

This research was funded in part by PROACTIVE 2017 "From Single-Cell to Multi-Cells Information Systems Analysis" (C92F17003530005 Department of Information Engineering, University of Padova) for BD Camillo; National Institutes of Health grant number U54CA21729 for J Ruan; Indian Council of Medical Research—Junior Research Fellowship for S Ahmad and X Wang was funded by the National Natural Science Foundation of China (No. 61702421 and No. 61772426). P Meyer thanks KV for help editing.

## Author Contributions

J Tanevski: conceptualization, data curation, software, investigation, methodology, and writing—original draft.
T Nguyen: software, formal analysis, and writing—review and editing.
B Truong: software, formal analysis, methodology, and writing—review and editing.
N Karaiskos: conceptualization, data curation, formal analysis, and writing—review and editing.
ME Ahsen: data curation, software, formal analysis, and writing—review and editing.
X Zhang: software, formal analysis, methodology, and writing—review and editing.
C Shu: software, formal analysis, methodology, and writing—review and editing.
K Xu: software, formal analysis, methodology, and writing—review and editing.
X Liang: software, formal analysis, methodology, and writing—review and editing.
Y Hu: software, formal analysis, methodology, and writing—review and editing.
HVV Pham: software, formal analysis, methodology, and writing—review and editing.
L Xiaomei: software, formal analysis, methodology, and writing—review and editing.
TD Le: software, formal analysis, methodology, and writing—review and editing.
AL Tarca: software, formal analysis, and methodology.
G Bhatti: software, formal analysis, methodology, and writing—review and editing.
R Romero: software, formal analysis, methodology, and writing—review and editing.
N Karathanasis: software, formal analysis, and methodology.
P Loher: software, formal analysis, methodology, and writing—review and editing.
Y Chen: software, formal analysis, methodology, and writing—review and editing.
Z Ouyang: software, formal analysis, methodology, and writing—review and editing.
D Mao: software, formal analysis, methodology, and writing—review and editing.
Y Zhang: software, formal analysis, methodology, and writing—review and editing.
M Zand: software, formal analysis, methodology, and writing—review and editing.
J Ruan: software, formal analysis, methodology, and writing—review and editing.

C Hafemeister: software, formal analysis, and methodology.

P Qiu: software, formal analysis, methodology, and writing—review and editing.

D Tran: data curation, software, methodology, and writing—review and editing.

T Nguyen: software, formal analysis, methodology, and writing—review and editing.

A Gabor: data curation, software, methodology, and writing—review and editing.

T Yu: software, formal analysis, methodology, and writing—review and editing.

J Guinney: software, formal analysis, methodology, and writing—review and editing.

E Glaab: software, formal analysis, methodology, and writing—original draft.

R Krause: software, formal analysis, methodology, and writing—review and editing.

P Banda: software, formal analysis, methodology, and writing—review and editing.

G Stolovitzky: conceptualization, formal analysis, and writing—review and editing.

N Rajewsky: conceptualization, formal analysis, and writing—review and editing.

J Saez-rodriguez: conceptualization, funding acquisition, and writing—review and editing.

P Meyer: conceptualization, formal analysis, supervision, investigation, methodology, and writing—original draft.

## Conflict of Interest Statement

The authors declare that they have no conflict of interest.

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
