## [Reviewer comments · Life Science Alliance]

Life Science Alliance

Gene selection for optimal prediction of cell position in tissues from single-cell transcriptomics

Jovan Tanevski, Thin Nguyen, Buu Truong, Nikos Karaiskos, Mehmet Eren Ahsen, Xinyu Zhang, Chang Shu, Ke Xu, Xiaoyu Liang, Ying Hu, Hoang Pham, Li Xiaomei, Thuc Le, Adi Tarca, Gaurav Bhatti, Roberto Romero, Nestoras Karathanasis, Phillipe Loher, Yang Chen, Zhengqing Ouyang, Disheng Mao, Yuping Zhang, Maryam Zand, Jianhua Ruan, Christoph Hafemeister, Peng Qiu, Duc Tran, Tin Nguyen, Attila Gabor, Thomas Yu, Justin Guinney, Enrico Glaab, Roland Krause, Peter Banda, Gustavo Stolovitzky, Nikolaus Rajewsky, Julio Saez-Rodriguez, and Pablo Meyer

DOI: <https://doi.org/10.26508/lsa.202000867>

Corresponding author(s): *Dr. Pablo Meyer (IBM)*

Review Timeline:

Submission Date:	2020-07-31
Editorial Decision:	2020-07-31
Revision Received:	2020-08-13
Editorial Decision:	2020-08-21
Revision Received:	2020-08-26
Accepted:	2020-08-31

Scientific Editor: Shachi Bhatt

Transaction Report:

Please note that the manuscript was previously reviewed at another journal and the reports were taken into account in the decision-making process at Life Science Alliance.

July 31, 2020

RE: Life Science Alliance Manuscript #LSA-2020-00867-T

Dr. Pablo Meyer
IBM
Health care and Life Sciences
IBM T.J Watson Research Center
Yorktown heights, New York 10598

Dear Dr. Meyer,

Thank you for submitting your manuscript entitled "Gene selection for optimal prediction of tissue cellular position from single-cell transcriptomics" that was reviewed at another journal. Based on the reviewer reports at hand, we would be happy to publish your paper in Life Science Alliance pending final revisions necessary to meet our formatting guidelines.

The manuscript can be published pending some text and other minor modifications. Specifically, the manuscript needs to be carefully re-written to make sure that all the information and findings are easily accessible to the readers. Moreover, we would ask you to provide clarifications and further information in order to address the concerns regarding the benchmarking and the gold standard dataset. A clear pipeline that can be readily used in future analyses should be provided. No further comparisons to additional methods or further analyses of datasets are required. Please also provide a point-by-point response addressing the reviewers' concerns and take care of the following formatting requirements:

- please check that the author order in our system matches the author order in your manuscript
- please upload your main and supplementary figures as single files and add a Figure Legend section to your manuscript, with both the main and supplementary figure legends
- please double-check your figure callouts and make sure that you have a callout for both your main and supplementary figures in the main manuscript text
- please upload your manuscript text as an editable doc file

A. FINAL FILES:

B. MANUSCRIPT ORGANIZATION AND FORMATTING:

Sincerely,

Reilly Lorenz
Editorial Office Life Science Alliance
Meyerhofstr. 1
69117 Heidelberg, Germany
t +49 6221 8891 414

e contact@life-science-alliance.org
www.life-science-alliance.org

Dear Editor :

Thank you for considering our manuscript entitled “Gene selection for optimal prediction of cellular position in tissues from single-cell transcriptomics data” for publication in Life Science Alliance. We have now heavily edited the text, reordered the figures and simplified the conclusions. We also here attach the response to reviewer’s requests. We hope you will find this satisfying, otherwise please let us know any further changes.

Response to reviewers (in blue)

Reviewer #1:

While we are happy that the authors have included an additional data set, we believe that the benchmark still presents quality issues, including the lack of gold standards for the location mapping and the lack of additional data sets that allow to generalize the results. In addition, we think that is key that the authors provide a way for users to apply the best performing method, as that the main goal of a benchmark is that users can find what are the best solutions to the problem to eventually apply it in their data. Otherwise, the manuscript does not provide significant added value, apart from a reanalysis of an existing data set without new insights. Furthermore, the quality of the manuscript is low and is hard to read, without including a proper introduction to the state-of-the-art, introducing results before describing the challenges and without covering all supplementary figures. Finally, the authors do not compare or mention why one would prefer to perform ISH rather than use current spatial transcriptomics methods.

We have rewritten the manuscript to make it more accessible and clarified the implementation of the 10 best methods. Still we think that its main result i.e optimal gene selection for ISH and cellular position prediction is of value.

1. The order in the text is not very clear in some cases. For example, the authors describe that the best solution is the WOC; without explaining first what are the goals in the challenge or briefly describing what the methods are based on or mention that the quality and reproducibility of the solutions is ensured without describing how (Lines 26-39). The introduction, apart from the first paragraph, describes results rather than the state-of-the-art.

The first 2 paragraphs of the introduction discuss state-of-the-art and motivation for the challenge. We have moved the DREAM-related introduction to the end and complemented the introduction.

2. The code is scattered through different repositories, with little documentation and hardly generalizable to be applied in other data sets. As in our previous revision, we believe that is key that the authors provide a pipeline for users to be able to utilize the best method in the benchmark in their own data set.

All the methods together with their descriptions are available in Table S3 either dockerized (2) or under easily to execute R(4), Python(2), Matlab (2) scripts. They are set to be implemented using a 10-fold CV scheme that was easily adapted from Drosophila to Zebrafish.

3. In the same line, some supplementary figures are not in order or not mentioned in the text (e.g Figure S6), jumping from Figure S4 (line 102) to Figure S13 (line 153).

We are sorry about this, it has now been corrected.

4. The 'ground-truth' location of the cells in the zebrafish and Drosophila embryo depends on the results of another clustering method, DistMap. This is not really a gold standard. As described in other single-cell benchmarks (Saelens et al., 2019 Nat Biotech), a gold standard is defined when labels are given independently from the expression matrix (such as from cell sorting, the origin of the sample, or cellular mixing); otherwise it is a silver standard (usually by clustering the expression values).

We thank the reviewer for this comment and have now renamed the reference a 'silver standard'

5. As we suggested before, using simulated or semi-simulated data (e.g., using publicly available spatial transcriptomics data sets, where gene patterns can be sampled) the benchmark would include a challenge with a real gold standard.

We thank the reviewer for this comment but this goes beyond the goals of the manuscript.

6. The WOC solution results are not shown for the zebrafish data set.

The purpose of using the zebrafish dataset was to show that gene selection features and predictions were robust, this is why we did not implement WOC solution.

7. Another data set including ISH and single cell RNA-seq data has been generated in the wing disc, with a new method to perform gene selection and gene expression prediction on the wing template (Bageritz et al., 2019; Nat Methods), could be used in the benchmark.

We thank the reviewer for this comment but this goes beyond the goals of the manuscript.

8. How do methods compare with state-of-the-art methods, such as NovoSparc (Nitzan et al., 2019) or the one used by Bageritz et al., 2019? Both methods have been applied to the Drosophila Embryo data set.

We thank the reviewer for this comment, we discussed the comparison to NovoSparc and Bageritz et al in the manuscript.

9. If the genes selected by participants are differentially expressed, couldn't differential expression be used as gene selection method? Is performance the same for the methods, or do gene selection as done by the methods perform better?

We thank the reviewer for this comment, as all but 3 selected genes were differentially expressed, we doubt that dGeX would be an effective tool for gene selection.

Reviewer #2:

I remain enthusiastic about this study. But, the writing is still very hard to follow despite my comments on this in reviewing the previous draft. There is a bit more attention to topic sentences, and conclusions are stated in most sections, but the prose in between is convoluted and riddled with unnecessary details. This is unfortunate because as currently written I doubt most readers will suffer through to the end despite the presence of valuable results.

While I include a few specific comments below, I still feel that nearly every paragraph would benefit from a careful re-write and attempt at simplification.

We thank the reviewer for this comment and have now edited the flow of the manuscript's prose.

This is a stylistic comment but I think it would be better to move much of the last paragraph of the introduction to the beginning of the results, essentially starting the results with a concise description of the overall challenge, the subchallenges, and scoring strategy. Similarly, I would prefer to see a little more intuitive description of the scoring metrics in the main results section, not just the methods.

We thank the reviewer for this comment and have implemented his suggestion.

60-61 ("We further show..") - this sentence appears grammatically incomplete

We have corrected the grammar in this sentence.

?131-138 faulty logic re machine learning?

We edited this paragraph to clarify its message.

211 - parameters for tSNE belong in methods, not results section. Similarly the k-means clustering description is too detailed, obscuring the key logic of the approach.

We moved the tSNE and k-means parameters and description to the methods.

301-303: Not at clear why this observation is interesting enough to include. This is a characteristic example of the issues with the writing in this manuscript.

We excluded this observation.

August 21, 2020

RE: Life Science Alliance Manuscript #LSA-2020-00867-TR

Author information redacted

Dear Dr. meyer,

Thank you for submitting your revised manuscript entitled "Gene selection for optimal prediction of tissue cellular position from single-cell transcriptomics". We would be happy to publish your paper in Life Science Alliance pending final revisions necessary to meet our formatting guidelines.

Along with the points listed below, please also correct the following in the revised submission,

- please make sure that the author list in your manuscript matches the author list in our system
- please add the figure legends for both the main and supplementary figures to the main manuscript text
- please provide your tables in editable doc or excel format
- please label your figure file with the corresponding figure number
- please double-check your figures and figure callouts (you have a callout for figure 4A in the main ms text but Figure 4 does not have a Panel A)
- please bold the panel labels in the figure legends (eg. (A) (B)...)
- we also encourage you to get the manuscript text edited by a native or professional English speaker to improve clarity

A. FINAL FILES:

-- Summary blurb (enter in submission system): A short text summarizing in a single sentence the study (max. 200 characters including spaces). This text is used in conjunction with the titles of papers, hence should be informative and complementary to the title. It should describe the context

and significance of the findings for a general readership; it should be written in the present tense and refer to the work in the third person. Author names should not be mentioned.

B. MANUSCRIPT ORGANIZATION AND FORMATTING:

Sincerely,

Shachi Bhatt
Executive Editor
Life Science Alliance
www.life-science-alliance.org

August 31, 2020

RE: Life Science Alliance Manuscript #LSA-2020-00867-TRR

Author information redacted

Dear Dr. meyer,

Thank you for submitting your Research Article entitled "Gene selection for optimal prediction of cell position in tissues from single-cell transcriptomics". It is a pleasure to let you know that your manuscript is now accepted for publication in Life Science Alliance.

DISTRIBUTION OF MATERIALS:

Congratulations on a very nice paper. I hope you found the review process to be constructive and are pleased with how the manuscript was handled editorially. We look forward to future exciting submissions from your lab.

Sincerely,

Shachi Bhatt, Ph.D.,
Executive Editor
Life Science Alliance

e contact@life-science-alliance.org
www.life-science-alliance.org